gender; humanitarian contexts; global mental health; healthcare workers; conflict

**Corresponding author:**
Nadeen Abujaber;
Email: nabujabe@tcd.ie

# Gender considerations for supportive supervision in humanitarian contexts: A qualitative study

Elizabeth O'Sullivan[1], Nadeen Abujaber[1] , Meg Ryan[1], Kelly A. McBride[2], Pia Tingsted Blum[2] and Frédérique Vallières[1]

[1]Trinity Centre for Global Health, Trinity College Dublin, Dublin, Ireland and [2]Reference Centre for Psychosocial Support, International Federation of Red Cross and Red Crescent Societies, Copenhagen, Denmark

## Abstract

Supportive supervision has been shown to improve mental health outcomes and job retention for mental health and psychosocial support (MHPSS) workers in humanitarian contexts. However, the impact of gender on supervision practices has been poorly evaluated and documented in international guidelines to date. To address this gap, qualitative interviews were conducted with 12 MHPSS staff working in diverse humanitarian contexts to identify key gender considerations in supportive supervision. Results show that gender in supervision is influenced by the context of MHPSS work; with culture, religion and gender roles identified as key elements. Participants discuss recruitment mechanisms, highlighting the unequal gender distribution and inequitable opportunities within MHPSS programming. The importance of addressing power dynamics impacted by gender and of ensuring the safety of women within supervision is also highlighted. Finally, participants discuss the gender differences across the various supervisory formats. Altogether, results indicate that gender has the potential to influence supportive supervision within MHPSS, and it is recommended that international guidelines account for nuances of gender in supportive supervision within humanitarian contexts.

## Impact statement

The effect of gender on supervision in humanitarian emergencies has been largely ignored in international guidelines. Our study shows key areas where gender impacts the provision of supportive supervision. Accounting for gender in supervisory practices can improve the quality and effectiveness of supervision itself, contributing to more motivated and confident staff and providing mental health and psychosocial support to those impacted by humanitarian emergencies.

## Introduction

Mental health and psychosocial support (MHPSS) workers are responsible for delivering essential care in humanitarian contexts and often find themselves working long hours in demanding conditions, far from their social supports (IFRC, 2012; Roome et al., 2014; Brooks et al., 2016). Over time, MHPSS staff and volunteers are exposed to chronic occupational trauma, increasing their risk for burnout, post-traumatic stress disorder (PTSD), anxiety and depression (Musa and Hamid, 2008; Connorton et al., 2012; Lopes Cardozo et al., 2013; Charlson et al., 2019).

Supportive supervision has been highlighted as protective for the mental health of MHPSS workers in this context (Lopes Cardozo et al., 2012; Aldamman et al., 2019). Supportive supervision is defined as a "safe, supportive, confidential and collaborative relationship between a supervisor and supervisee, where supervisees can voice their difficulties, discuss challenges and be recognised for their successes, receive constructive feedback and emotional support, and build their technical skills and capacity" (McBride and Travers, 2021, p. 22). Supportive supervision acts in contrast to more traditional models of supervision, which are often seen as a managerial relationship overseeing employee performance (Clements et al., 2007; Coyle et al., 2022). Accordingly, high-quality, ongoing supervision has been shown to improve service delivery, enhance motivation and work satisfaction and decrease rates of staff turnover (Ndima et al., 2015; Vallières et al., 2018).

Despite the recognised benefits of supportive supervision, multiple studies have shown supervision to be poorly implemented, irregular and often absent in MHPSS programming within humanitarian emergencies (Crigler et al., 2013; Hill et al., 2014). Furthermore, there is also a paucity of standardised guidelines for supportive supervision in MHPSS activities in

humanitarian emergencies (Raven et al., 2020). The World Health Organisation published guidelines for supportive supervision but it was focused on mid-level managers in routine health encounters, not humanitarian contexts (WHO, 2008). Abujaber et al. (2022) conducted a systematic scoping review to examine empirically supported supportive supervision practices in humanitarian contexts but this was not specific to MHPSS activities. Addressing the need for standardised guidelines on supportive supervision in MHPSS in humanitarian contexts, "Supervision: The Missing Link" project was launched in 2019 as a collaboration between the Trinity Centre for Global Health (TCGH), Trinity College Dublin and the International Federation of Red Cross and Red Crescent Societies Psychosocial Centre (PS-IFRC). The project used participatory research approaches spanning multiple stages of stakeholder consultation, including a desk review, key informant interviews (Perera et al., 2021), and a Delphi consultation (Travers et al., 2022), to inform the development of a (freely available) Integrated Model for Supervision (IMS) Handbook and accompanying training guidelines (http://www.supervision-mhpss.org/).

However, existing international guidelines for MHPSS programming in the humanitarian sector contain minimal information about the impact of gender on supportive supervision. In the guiding frameworks for MHPSS programming within emergency contexts published by both the Deutsche Gesellschaft für Internationale Zusammenarbeit GmbH (GIZ) and the Inter-Agency Standing Committee (IASC), gender is addressed in the following ways: the difference in psychological distress reactions of men and women; the need for different treatment strategies for men and women; the recommendation to use same-gender interviewing for sensitive discussions (IASC, 2007, 2018), and recognising that participation may be hindered by gender roles (German Cooperation, 2018). Notably absent – from these guidelines and the extant literature – is an examination of how gender may impact supportive supervision and how gender should be addressed in supervisory settings. This represents an important gap, given what we currently know, that humanitarian programmes which assess and address gender-specific issues, enhance agency and leadership of women, and promote gender equality, yield more effective and equitable results (Lafrenière et al., 2019). To address the noted absence of guidance for gender considerations within supportive supervision for MHPSS programming, the current study had two objectives: (i) to identify key gender considerations for supportive supervision within MHPSS programming as well as to (ii) ascertain how gender considerations should be incorporated into the IMS Handbook and accompanying training guides.

## Methods

### *Study design, participants and procedures*

This study employed a qualitative methodology using semi-structured key informant interviews to explore the views and experiences of MHPSS workers regarding the impact of gender and gender roles on supportive supervision in humanitarian settings. Interviews were also designed to elicit recommendations from participants who had received training in the IMS approach regarding areas within the IMS guidelines where gender considerations could be applied and strengthened.

MHPSS practitioners with experience either as managers, supervisors and/or supervisees were recruited using purposive sampling. Participants who were selected worked with humanitarian organisations in various contexts to gain as rich and as diverse a

perspective as possible. Identification of potential participants was performed by the gatekeeper (FV), an experienced MHPSS expert and researcher, as well as through participants involved in IMS training and implementation via the Missing Link Project. IMS Training consisted of operationalising the "IMS Handbook" into a training curriculum consisting of a series of four training modules, bookended by a pre-training meeting and follow-up supervisory support and implementation consultations, which was delivered in line with the apprenticeship training model (Murray et al., 2011). A total of 90 individuals took part in the IMS Training held between June and July 2021. All IMS Trainings were conducted in English and consisted of two training cohorts, the first representing organisations in Nigeria and Jordan and the second from Afghanistan and Ukraine. The trainings were held online due to the COVID-19 pandemic restrictions.

A total of 12 ($N = 12$) interviews were conducted with experienced MHPSS workers. Seven (women = 4; men = 3) of these participants, identified as participants A to G in Table 1, were interviewed by the first author (EO) between September 2020 and January 2021 (with delays due to the COVID-19 pandemic) and the five remaining participants (women = 4; men = 1; identified as participants H to L in Table 1 who were involved in the IMS training between June and September 2022) were interviewed by the second (NA) and third (MR) authors. Participants interviewed by EO had never met her prior to the interview nor did they have any knowledge about her background apart from being a Masters student. Participants interviewed by NA and MR were aware of their backgrounds as researchers and their involvement in the IMS trainings though they were assured of their anonymity and confidentiality. The interviews lasted between 30 and 60 min. At the time of the interview, participants were working with eight different humanitarian organisations, from nine countries of origin (Jordan = 2; Ukraine = 2; Afghanistan = 1; USA = 2; Italy = 1; Spain = 1; Netherlands = 1; Australia = 1; Syria =1). Five participants were in management roles, seven acted as supervisors while also receiving supervision, but none were solely supervisees. Two participants held roles as technical advisors and one held a training role alongside their work as supervisors. Participant details are summarised in Table 1.

**Table 1.** Participant details

| Participant code | Gender | Role | IMS training attendance | Interviewer |
|---|---|---|---|---|
| A | W | Supervisor/Supervisee | No | EO |
| B | M | Supervisor/Supervisee | No | EO |
| C | W | Supervisor/Supervisee | No | EO |
| D | M | Management | No | EO |
| E | M | Management | No | EO |
| F | W | Supervisor/Supervisee | No | EO |
| G | W | Supervisor/Supervisee | No | EO |
| H | M | Management | Yes | MR |
| I | W | Supervisor/Supervisee | Yes | NA |
| J | W | Supervisor/Supervisee | Yes | NA |
| K | W | Management | Yes | NA |
| L | W | Management | Yes | MR |

EO, Elizabeth O'Sullivan; M, Man; MR, Meg Ryan; NA, Nadeen Abujaber; W, Woman.

## Data collection

EO contacted all the potential participants identified by FV via email and NA and MR emailed all those who had been involved in the IMS training, providing them with the participant information leaflets and informed consent forms. Those who expressed interest in participating were scheduled for interviews. Interviews were conducted using Zoom Video Communications (2020) and stored on a secure server, accessible only to the research team, in a password-protected file. Data was pseudo-anonymised at the point of transcription through the redaction of names of individuals, specific locations, humanitarian organisations, and job titles, with each participant represented by a unique code. Participants had the right to review their transcripts upon request and make any changes up until November 2022.

Semi-structured interview questions (see Supplementary Material) explored a wide range of supervisor elements potentially impacted by gender and gender roles, including supervisory pairings and relationships, supervision formats (individual, group, and peer), the ability to perform supervisory tasks, access to supervision training, and the recruitment of supervisors. For participants who had taken part in the IMS training, questions also evaluated areas within the IMS guidelines where gender considerations could be applied and strengthened.

## Data analysis

The audio recordings were transcribed into text verbatim. Transcriptions were then checked for accuracy by listening to the audio files while reading the corresponding transcripts and correcting them accordingly. All interviews were conducted in English, in which all participants and the researchers were fluent. Words spoken in Arabic were translated to English, by NA, as a fluent English and Arabic speaker, directly during transcription.

A thematic analysis framework, as proposed by Braun and Clarke's (2006) six-step approach, was used to analyse the qualitative data. First, the researchers became familiar with the data by reading the transcripts while simultaneously listening to the audio files. Next, the entire transcript was examined systematically to identify codes. EO used NVIVO 12 Pro (QSR International, 2022) for data analysis while NA and MR conducted their analysis manually. Similar codes were then grouped into subthemes by the researchers who then searched for commonalities amongst these subthemes to create themes. Once themes were identified, they were reviewed by the researchers to ensure they covered the breadth of topics discussed by participants. Themes and subthemes generated by EO were then compared to those generated by NA and MR to identify overlapping topics and contrasting ideas. As a final step in this analysis, the finalised list of themes and subthemes was applied as a gender framework to the IMS Handbook to identify gaps and areas in need of strengthening. NA and MR independently evaluated the IMS Handbook and compared results to increase reliability.

## Results

### Thematic analysis of qualitative interviews

Themes, subthemes, and codes are summarised in Table 2. Items in bold depict codes with the strongest overlap between coders.

**Table 2.** Combined themes, subthemes, and codes from first, second and third authors

| Themes | Subthemes | Codes |
|---|---|---|
| Context | A Culture | **Acceptability of mixed-gender meetings and trainings** |
| | | Same-gender protection |
| | B Religion | Christianity |
| | | Islam |
| | C Gender roles | Household duties |
| | | **Power imbalance** |
| Recruitment | A Numbers of women and men | **More women in MHPSS** |
| | | **More men in senior positions** |
| | B Inequitable opportunities | **Lack of education opportunities for women** |
| | | **Women having less experience in the field** |
| Role of the supervisor | A Addressing gender | Reflective practice |
| | B Safety of women | Unsafe areas |
| | | Sexual harassment |
| | | Coordinate with local community for safety |
| Supervision formats | A Individual | **Ideally same-gender (for emotional support)** |
| | | Influence of local context on emotional topics |
| | B Group | Same gender preference |
| | | Gender of less Importance |
| | C Remote | More open to sharing |
| | | Pressures at home |

### Context

All participants mentioned that gender considerations for MHPSS supervision would greatly depend on the context of their work. Context included the subthemes of culture, religion, and gender roles.

A. *Culture*: The cultural and societal norms of an area can impact how supervision models are created. As Participant C (supervisor/supervisee) put it, "*Culture is key in the issue of gender.*" Many participants described situations where due to the local culture in certain contexts, it would not be appropriate to have mixed-gender supervision.

> Can I actually use a male to supervise? Because if they are going to do live supervision, they need to go into a room where there are only 2 females, and that client is not going to be comfortable to talk with a male in the room so how do they provide supervision? – Participant L (Management)

Not only did gender impact supervisory pairings, but Participant L also discussed the challenges of mixed-gender trainings in more conservative contexts: "*we are having difficulties with training colleagues of the different genders within the same sessions.*" These restrictions lead to suboptimal training experiences as she further describes:

They [participants] were in a venue and there was a curtain down the middle, with males on one side and females on the other side. And the facilitator was told that he must not go over to the female side. So, I'm like: 'how do we know what the females are doing? How do we know that they are understanding the training?'

In addition to the importance of same-gender supervisory pairings in certain contexts, one participant further described an element of protection in having supervisors and supervisees of the same gender within traditional and conservative settings.

We try to avoid having a male and female alone, it's not even if they would be very comfortable, that's also just to avoid any misconceptions, it's to protect both of them. – Participant F (Supervisor/ Supervisee)

By contrast, one participant felt that the organisational culture superseded local culture in terms of gender considerations for supervision. She reported that, even in more conservative cultures, "*within the international NGOs, it is very sensitive towards gender things. I don't believe these (gender) issues have happened.*" – Participant I (Supervisor/Supervisee)

B. *Religion:* Gender expectations can be greatly influenced by the religion of a population. Participant E (Supervisor/Supervisee) noted a difference between Muslim and Christian communities within one country: "*There are both Christians and Muslims. With the Christians it was fine. But with the Muslims, they definitely…prefer if it's one-on-one supervision, to be of the same gender.*" Participant J (Supervisor/Supervisee) mentioned the link between religion, gender roles and gender mixing but felt that in contexts where "*religion is divided from the government…still our society supports these gender roles,*" alluding to the influence of culture on gender norms in less religious contexts.

C. *Gender roles:* Nearly all participants reflected on how gender roles in certain contexts have hindered women from completing their MHPSS tasks and supervisory activities because women are still expected to take care of the household and children, while simultaneously navigating their work responsibilities. As reported by participant A, a technical advisor who is both supervisor and supervisee: "*They've got kids at home, they've got extended family that they're looking after, they're expected to cook every meal, they're expected to take care of their children.*"

Not only did fulfilling these traditional roles tend to interfere with completing MHPSS and supervisory tasks, but participants noted that it also generates a power imbalance impacting career options. For example, Participant J (Supervisor/Supervisee) noted: "*I think it is gender specialisation. Men have to be the hero but for women, they cannot act like this,*" emphasising the stigma and backlash faced by women who try to move beyond the rigidly prescribed roles for their gender. In addition, Participant E (Supervisor/Supervisee) highlighted that some women have been forced to step back from their career and supervisory roles to focus on their family duties, fulfilling gender role expectations that they did not have the power to negotiate: "*Some having to drop out from the project, one had to take a reduced caseload because she had to take care of her children more than her husband even though her husband didn't really have a job.*" These power dynamics were also felt to impact the ability of women to conduct their supervisory roles because "*men are generally seen as stronger and right so unless you have a confident and competent woman, a lot of them (women) are going to say: 'yes, no, you are right' without being able to have that collaborative relationship.*" Participant L (Management)

## Recruitment

Another recurring theme was the issue of recruitment of both supervisors and supervisees within MHPSS, with a focus on the following subthemes: gender distribution in MHPSS and the inequity of recruitment opportunities.

A. *Gender distribution:* Most participants reported that the number of women in MHPSS far outweighs that of men, with Participant C (Supervisor/Supervisee) noting that "*MHPSS is very much female-dominated.*" Participant H (Management) confirmed this stating that there are more women candidates for MHPSS positions within their context: "*most of the mental health specialty here is preferred by females so we have the vast majority of, for example, psychological counselling, psychologist, nurses, pharmacists, are females so that could explain the gender representation in our agency.*" By contrast, Participant L (Management) reported that their "*staff is quite male dominant*" with efforts made to "*grow the number of females.*" She emphasised the importance of gender balance in recruitment and training, hoping that during IMS implementation in their context, "*especially on the supervisor level, to make sure to have enough males to support the males and enough females to supervise the females.*"

However, though the majority of participants found that women made up much of the MHPSS workforce in terms of supervisees, they noted that most leadership positions, those with higher authority and power, were occupied by men. In Participant G's (Supervisor/Supervisee) words: "*If we're talking about numbers, it's dominated by females, but for talking about positions and seniority, males dominate that.*" In contrast, Participant K (management) denied "*preferences for men in management positions. It usually depends on skills and experiences*" though she admitted that men tended to be less interested in MHPSS and supervision in general compared to women.

B. *Inequitable opportunities:* Many participants noted that the opportunities afforded to women and men are heavily influenced by the culture in which they live. In many contexts where MHPSS services are provided, women have fewer educational and career opportunities due to their gender and the gender norms of their society, particularly in conservative countries with more traditional gender roles. As reported by Participant G (Supervisor/Supervisee): "*Because women have less opportunities, they're often less technically skilled, or they have less education than men here.*" However, this participant affirmed that this should not preclude women from progressing in MHPSS and urged humanitarian organisations to address these historic injustices and inequitable opportunities for women during the recruitment process "*to make sure that we have a spread of different people so that all supervisors aren't men.*" Participant G

Participant L (Management) describes the active efforts made by her organisation to include, recruit and train women as follows:

We have discussions with female staff to understand what will help them apply for jobs, how do we promote them, how do we capacity build them? We have gotten quite a push recently, we have a mentorship program with them, any capacity building we are doing with them, as well as for recruitment: we say it is a female only position because it makes them feel more confident to apply.

### Role of the supervisor

Participants also stressed the importance of addressing gender dynamics with their supervisees and protecting the safety of women supervisees

A. *Addressing gender:* Participants reported that it is a vital role of the supervisor to address power dynamics, including gender, with their supervisees, highlighting the importance of reflective practice as a supervisor.

> What makes it successful is that the supervisor is able to reflect on their position of power, their gender, how that might influence a relationship, so that they can deconstruct that with their supervisee. – Participant G (Supervisor/Supervisee)

B. *Safety of women supervisees:* Be it living in an area of conflict or overcrowded camps, MHPSS workers and local volunteers are at risk of violence. Participants noted that the women working for MHPSS programs in these situations are at higher risk than men, and logistics often need to consider gender:

> MHPSS practitioners, particularly females who are working in refugee camps or in fragile, unsafe contexts are more prone to sexual harassment and other kinds of abuse from the hostilities, from the communities that they are serving because they are young females providing those activities. – Participant A (Supervisor/Supervisee)

Participants highlighted a key responsibility of a supervisor was working with the local staff and community to identify dangerous areas for women to help ensure their safety.

### Supervision formats

Participants also noted differences between gender considerations in individual, group, and remote supervision.

A. *Individual supervision:* Participants reported that individual supervision tends to be easier when the gender of the supervisor and supervisee is the same, as there is a level of common ground and understanding to start with. Participants felt this was especially relevant when a supervision session centred on emotional support. As stated by participant G (supervisor/supervisee): "*It's a bit easier with someone who's the same gender as you. Because that's often where a lot of the deep things come up.*" Participants highlighted the influence of local norms and culture regarding mental health and emotional support on gender supervisory pairings.

> I think when we say emotional support, really depends on the cultural norms and acceptance. Usually, it would be more accepted for men to receive emotional support from men, and for females to receive and accept emotional support from females… usually, the males can understand each other much better, they have mutual understanding as they live the same experience, and the way they interact and what kind of support they may look for. – Participant H (Management)

B. *Group supervision:* Most participants stated that they found same-gender groups for supervision were more beneficial for their supervisees in humanitarian contexts. They felt that when in a group of same-gender peers, women and men felt they were able to speak more freely. "*Even if they say it's okay to mix, I felt women tend to be more free in what they say and do when they're only females and as soon as males walk in it's different.*" Participant F (Supervisor/Supervisee). Only one participant offered a different perspective, that the gender of the supervisor and supervisee matters less in group supervision as the discussions tended to be less personal.

C. *Remote supervision:* One participant observed that it can be more comfortable for women in certain contexts to be able to participate in supervision from the comfort of their own home.

> A lot of people who weren't disclosing very much, were being really vulnerable and honest in their online supervision, because they can have their camera off, they can be where they want to in their house, they don't have to see my reaction if they want to tell me something really personal… And it can be more comfortable for women. – Participant G (Supervisor/Supervisee)

However, while remote supervision may allow women supervisees to feel more comfortable, some participants reported the added burden for some women to complete all their household work while also working remotely given the gender roles of their culture.

> If women are at home and also supposed to be working, they have the double burden of the family expects them to be available and still cooking and still looking after children and work expects them to be on zoom and doing all their work tasks as well. So, I had some of the women telling me how hard it was. – Participant G (Supervisor/Supervisee)

### IMS handbook

The themes and subthemes from Table 2 were subsequently applied as a gender framework to the IMS Handbook in a document analysis. The following gender elements were found in the IMS Handbook: contextual factors impacting mixed-gender supervisory pairings and the impact of gender in power dynamics and boundary setting in supervision. Notable gaps in the IMS Handbook regarding gender considerations included the impact of religion and gender roles on supervision, organisational responsibility to provide equitable training and professional development opportunities for both genders including the promotion of women to supervisory positions, the role of the supervisor in addressing gender and ensuring the safety of women supervisees, as well as the impact of gender on different supervisory formats.

### Discussion

This study aimed to identify key gender considerations for MHPSS supervision within humanitarian emergencies and to apply these considerations to the IMS Handbook to strengthen its application of gender in supervisory practices. Participants reported that gender was considered particularly important for the following elements of MHPSS supervision: the context in which supervision was taking place, the supervisory pairing, recruitment opportunities in supervision, the role of the supervisor, and supervisory formats.

Results indicate that a one-size-fits-all approach regarding gender is insufficient for MHPSS supervision, as the cultural, religious and community influences need to be taken into consideration to ensure that supervision is safe, effective, and accepted by the local community. A recent study found that mental health practitioners working in emergency contexts considered that best practice for MHPSS supervision should "incorporate awareness of the relevant cultures or contexts" (Perera et al., 2021). Supporting the findings of Perera et al. (2021), the results of the current study reinforce the importance of maintaining an awareness of the local religious and cultural contexts and their influence on gender roles and the perceived appropriateness of mixed-gender supervision. Participants reported that in certain contexts men supervisees were less comfortable accepting feedback and emotional support from women, due to socio-cultural norms where men were expected to hold positions of power. This supports the findings of Crigler et al.

(2013) who propose that maintaining an awareness of power dynamics regarding gender is important for supervision, as the management of power hierarchies is key to creating a safe, trusting, and transparent supervisory relationship (Thomas et al., 2019). Such imbalances in MHPSS work were reported by participants, who stated that while MHPSS is a field with more women working in it, the majority of those in positions of power were men. For example, in our sample, all men included were managers or supervisors in leadership positions. Gender roles were noted in this study to hinder women's ability to progress to supervisory roles due to time constraints and domestic responsibilities. Other factors like lack of education opportunities also make women less likely to be selected for supervisory roles. While further research is needed to quantify the numbers of women and men in supervisor and supervisee roles, MHPSS organisations should examine their own practices and hiring mechanisms to ensure that they are giving equal opportunities to women and men and eliminating the structural barriers contributing to the unequal gender distribution in MHPSS workforces.

Results highlight the importance of teaching supervisors to reflect on their preconceptions regarding gender and how societal expectations may influence their supervisory relationships. Participants endorsed that supervisors must be self-reflective and taught how to address gender with their supervisees in a culturally appropriate manner. While the importance of supervisors addressing power dynamics and gender in the supervisory relationship has been seen in studies with trainee counsellors (Walker et al., 2007; Thomas et al., 2019), this is the first study to extend these results to MHPSS supervision in humanitarian contexts.

Gender considerations also differ depending on the supervision format being used. Most participants felt supervisees were more comfortable with a same-gendered supervisor in individual sessions in certain contexts, while for group supervision, the results were mixed as to whether same or mixed genders were preferred. While studies have discussed the impact of gender pairings on individual supervision (Van der Veer et al., 2004; McBride and Travers, 2021), to date, data on group supervision has focused on group dynamics but not on how gender impacts group supervision specifically. Given the inconclusive nature of this study's results on the interaction between gender and group supervision, further research is warranted. For remote supervision, results indicated that this type of supervision can facilitate an increased sense of safety for supervisees, especially women operating in dangerous contexts. However, results also noted the difficulties in obtaining the privacy and protected time needed for effective supervision due to gendered household responsibilities. Further research on strategies to enhance the effectiveness of remote supervision for all genders would be beneficial given the increasing popularity of this supervision modality in recent years.

Altogether, results indicate that gender has the potential to influence the process and outcome of supportive supervision within MHPSS, and it is suggested that international guidelines are re-examined and updated to advocate for an awareness of the nuances of gender in supportive supervision in humanitarian contexts.

### Implications for the IMS

This study highlights an existing gap in gender considerations within formal, international supervisory guidelines for humanitarian contexts, including the IMS Handbook. Applying a gender lens during the ongoing development of the IMS guidelines and incorporating the specific topics featured in this qualitative study as part of future versions of the IMS is necessary to make it as gender transformative as possible. This includes but is not limited to: the choice of supervisory pairings and formats as influenced by local context, culture and religion, the training of supervisors about the impact of gender on power dynamics with supervisees, as well as the recommendations for organisations to provide equitable access for all genders to education, career development and promotion to supervisory roles, taking into account the added responsibilities often faced by women as dictated by gender roles in different contexts. It is also recommended that gender balance be obtained, where possible, when recruiting for training on the IMS, and when recruiting research participants to reflect on the acceptability and utility of the IMS.

### Limitations and considerations for future research

The current study is not without limitations. Most apparent are the imbalances regarding gender inclusion, with twice as many women as men taking part. However, this may be due to the context of MHPSS work, noted by participants as a field with higher numbers of woman. Although 90 individuals took part in the IMS training, there were several barriers to participation in interviews. Due to the COVID-19 pandemic, interviews were conducted online, which could have excluded participants with poor internet access. At the time of data collection, Ukraine and Afghanistan were also dealing with humanitarian crises, likely impacting research participation. Interviews were conducted in English which may have acted as a barrier for non-native speakers. Furthermore, though complete confidentiality and anonymity were assured, the opinions expressed by participants who had engaged in IMS training may have been impacted by the involvement of NA and MR in the IMS project. Finally, the current study is limited by a lack of supervisees, which may impact the results as each cohort may have differing views on how gender impacts supervision. Future research should focus on the experiences of supervisees to better understand gender and supervision in MHPSS.

### Conclusion

Effective supportive supervision is essential to the provision of MHPSS services in humanitarian crises. This study highlights some of the important gender considerations in supportive supervision, with the hopes of strengthening gender considerations within international guidelines (such as the IMS) and better-integrating gender into supportive supervision.

**Open peer review.** To view the open peer review materials for this article, please visit http://doi.org/10.1017/gmh.2023.33.

**Supplementary material.** The supplementary material for this article can be found at https://doi.org/10.1017/gmh.2023.33.

**Data availability statement.** The data supporting this study contain potentially identifiable information and is not publicly available for data protection reasons. Data can be made available by the author upon reasonable request.

**Acknowledgements.** Many thanks to our participants for their time and valuable contributions to our project.

**Author contribution.** E.O., N.A., M.R.: Study design, data collection, data analysis, interpretation of results, writing and drafting. K.A.M., P.T.B.: Project administration, writing – reviewing, editing and final approval. F.V.: Study

design, supervision, project administration and management, writing – reviewing, editing and final approval.

**Financial support.** This study is made possible by the support of the American People through the United States Agency for International Development (USAID; Grant #: 720FDA19IO00106). The contents of this study are the sole responsibility of the authors and do not necessarily reflect the views of USAID or the United States Government.

**Competing interest.** The authors have declared that no competing interests exist.

**Ethical statement.** This study forms part of the Missing Link project, which was granted ethical approval from the Trinity College Dublin Health Policy and Management/Centre for Global Health Ethics Committee on January 17, 2020. All participants provided written consent in English prior to their interviews. They were also informed of their right to withdraw from the study at any time. Participants provided their approval for the use of their anonymous quotes prior to publication where applicable. Ethical standards were followed to ensure confidentiality and anonymity of the collected data.

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
