## [Reviewer Report]

Dear Professors Bass and Chibanda, 

We are pleased to submit our manuscript ‘’Gender considerations for supportive supervision in humanitarian emergencies: A qualitative study” to be considered for publication in Cambridge Prisms: Global Mental Health. 

Supportive supervision has been shown to be a protective factor in the mental health of MHPSS workers in humanitarian contexts. However, there is a paucity of standardized guidelines for supportive supervision in humanitarian emergencies. Moreover, there is minimal information in international literature regarding the impact of gender on supportive supervision in humanitarian contexts. To address the gap in supervision guidelines, a project was launched by the Trinity Centre for Global Health, Trinity College Dublin and the International Federation of the Red Cross and Red Crescent Societies Reference Centre for Psychosocial Support to codevelop and create the ‘Integrated Model for Supervision’ (IMS). To ensure that the IMS is as gender transformative as possible, this study aims to identify key gender considerations in supportive supervision within MHPSS programming in humanitarian contexts and determine how best to incorporate these gender elements within the IMS and its supplemental materials. 

Results highlight the following key areas where gender impacts supportive supervision within MHPSS programming: the context in which supervision is taking place, the supervisory pairing, recruitment and career development opportunities in supervision, the role of the supervisor, and supervisory formats. Altogether, results indicate that gender has the potential to influence the process and outcome of supportive supervision within MHPSS, and it is recommended that international guidelines, including the IMS, are re-examined and updated to reflect the nuances of gender in supportive supervision in humanitarian contexts. 

The manuscript is an original piece of research and has been prepared in accordance with the journal style. The manuscript is 4571 words long (excluding abstract, references, and tables). The manuscript has not previously been published and is not under consideration for publication elsewhere. I have assumed the role as corresponding author and all co-authors have agreed with the order of the author list. 

Thank you in advance for your kind consideration. 

Regards,

Nadeen Abujaber, MD

Trinity Centre for Global Health, Trinity College Dublin

---

## [Reviewer Report]

General feedback:

The study highlights some important gender considerations in supportive supervision for MHPSS within humanitarian emergencies and the authors make some propositions to strengthen its application.

Below are comments and observations which, I believe, will help the authors to improve their manuscript.

The authors should check the whole manuscript for any grammatical errors.

Introduction: Generally, this section is well written. However, the authors should include the definition of supportive supervision on page 2, between lines 7 and 8.

Methods:

• Page 3, Line 55 -include the word ‘supportive’ before supervision. This should read ‘supportive supervision’ to differentiate it from the other supervision formats.

• Page 3, line 60 – there is a mix-up of words which makes the sentence unclear – I believe it should read ‘Participants who were selected worked….’ and not ‘Participants were selected who worked….’ Please review this sentence for clarity.

• Page 4, Lines 83 - 86 - The authors should mention who conducted the interviews. What did the participants know about the researchers? It is important to reflect on this as it might introduce some biases.

• Under the data analysis section, page 5, lines 107 to 112, the authors should indicate how the coding was done i.e., was any software used for coding? If yes, please include the name of the software used.

• At what level was the saturation point reached? It would be important for the authors to include this in their methodology section.

• I suggest that the authors include the semi-structured interview guide used as an additional/supplementary file to this manuscript.

Results: The authors have presented their results very well and it is easy to read and follow through the thematic areas. The use of ‘quotes’ from the interviews is indeed useful in getting the voice of the respondents.

Discussion and Conclusion: The authors have discussed their findings logically and concisely and they have pointed out areas that require further research. The limitations of the study are acknowledged except for the reflexivity bit which is lacking.

This research is an important contribution to the general body of knowledge in this field. I hope the comments will be helpful and motivate the authors to improve the manuscript.

Good luck with this piece of work.

---

## [Reviewer Report]

Thank you for the opportunity to review this manuscript. The authors have used key informant interviews to explore a key (and largely ignored) issue: the role of gender in supervision of task-shared MHPSS programs. They interviewed a group of mostly senior MHPSS managers and supervisors and did not interview any exclusive supervisees. The manuscript is strong, well-written, concise, and clear, and the results will be directly applicable to the field and to revisions to the IMS guidelines. I have only minor comments.

- Can you clarify the difference between the five IMS training participants and the other seven, who you describe as experienced MHPSS workers? Are the five not currently MHPSS workers? In addition, please consider adding IMS attendance as a variable to Table 1. I struggled to understand how many of your participants had not attended IMS training but were nevertheless in MHPSS provider roles.

- Operationalizing these recommendations will have substantive implications for the MHPSS workforce in terms of its gender distribution. You note this, but do not explicitly discuss whether this distribution is realistic. Please consider expanding your discussion to include (any available) statistics on the gender distribution of the MHPSS workforce (by supervisor/supervisee) and indicate whether structural efforts would need to be made to improve gender alignment in supervisor/supervisee relationships.

- I noted that your themes and subsequent recommendations are basically all structural in nature (related to the gender identity of the supervisor and supervisee), but that there were no themes related to differences in preferences for supervisory approach and content by gender. Did different gender participants express different preferences around the content of supervision? For example, that supervision should be on balance positive/reinforcing vs. punitive or critical, that it should be more structured vs. unstructured, or that recording devices and other tools should or should not be used? If so, please consider expanding the results to discuss these issues.

---

## [Reviewer Report]

This is a very well written article addressing a rarely considered aspect of MHPSS and supervision. The research topic and methods are sufficiently clear. The results and discussion are, in particular, extremely well-presented, with conclusions from the article definitive for future action(s).

The priority revision I suggest relates to Table 1 and the participants information. Table 1 - Participant details would benefit from including the country from which the participants are from. When reading the results, many views seemed gender-conservative and I kept wondering if the sample was skewed. I could not ascertain this because the participants are said to be from four different countries, but the breakdown of which countries (by participants) is not provided. If there are anonymity concerns for including countries in Table 1, the methods should, at a minimum, list how many of the 12 participants are from which of the four countries. If there was a skew in country representation from participants, this might also need to be reflected in the study limitations.

Some additional suggestions below could further strengthen the article, although I would not consider these issues a barrier to publication. I encourage the authors to consider the following:

*Only one supervision guidance document is cited in the research, when there are now several available. A few lines outlining these could demonstrate both paucity but also growing interest in the value/importance of supervision in MHPSS humanitarian work.

*References citing wellbeing of humanitarian workers is all very dated and more recent references would be ideal

*Line 23 - you mention international guidelines for MHPSS programming in fragile states, but you could widen this to the overall humanitarian sector

*Mention and introduction to the IMS Handbook earlier in the introduction section would have better tied in the focus and specifics of the research - this is linked to my next point....

*Paragraph from lines 25-37 took me some time to comprehend, because the content seemed to conflate MHPSS guidance, MHPSS gender guidance, supervision guidance and supervision guidance with gender considerations. My first reading made me unsure if you were address MHPSS generally or supervision specifically. It eventually became clearer after reading the next paragraph (beginning line 38). However, perhaps a restructuring of this paragraph could make for easier reading, especially for non-English-as-first-language speakers.

*Line 63 - recommend that you initial which author is the “gatekeeper”. Other references to authors roles in the analyses are described as first, second, third author, etc. This is acceptable, but is often easier for the reader if these are also initialed.

*In the methods, only 5 out of the 90 IMS-trained participants were included in the sample. It would be useful to know how these were selected (such a small number from such a large cohort)

*The thematic constructs of “Gender roles” as a sub-theme and the code of “Power Imbalance” seemed to blend in the results (e.g., Paragraph beginning line 169-184). There may be value in questioning whether “power imbalance” warranted being its own sub-theme rather than a code of gender roles.

My congratulations to the authors of this excellent article addressing an important aspect of MHPSS in emergencies work.

---

## [Reviewer Report]

Dear Professor Chibanda, 

We are pleased to submit our manuscript ‘’Gender considerations for supportive supervision in humanitarian emergencies: A qualitative study” to be considered for publication in Cambridge Prisms: Global Mental Health. 

Supportive supervision has been shown to be a protective factor in the mental health of MHPSS workers in humanitarian contexts. However, there is a paucity of standardized guidelines for supportive supervision in humanitarian emergencies. Moreover, there is minimal information in international literature regarding the impact of gender on supportive supervision in humanitarian contexts. To address the gap in supervision guidelines, a project was launched by the Trinity Centre for Global Health, Trinity College Dublin and the International Federation of the Red Cross and Red Crescent Societies Reference Centre for Psychosocial Support to codevelop and create the ‘Integrated Model for Supervision’ (IMS). To ensure that the IMS is as gender transformative as possible, this study aims to identify key gender considerations in supportive supervision within MHPSS programming in humanitarian contexts and determine how best to incorporate these gender elements within the IMS and its supplemental materials. 

Results highlight the following key areas where gender impacts supportive supervision within MHPSS programming: the context in which supervision is taking place, the supervisory pairing, recruitment and career development opportunities in supervision, the role of the supervisor, and supervisory formats. Altogether, results indicate that gender has the potential to influence the process and outcome of supportive supervision within MHPSS, and it is recommended that international guidelines, including the IMS, are re-examined and updated to reflect the nuances of gender in supportive supervision in humanitarian contexts

The manuscript is an original piece of research and has been prepared in accordance with the journal style. The manuscript is 4998 words long (excluding abstract, references, and tables). The manuscript has not previously been published and is not under consideration for publication elsewhere. I have assumed the role as corresponding author and all co-authors have agreed with the order of the author list. 

Thank you in advance for your kind consideration. 

Regards,

Nadeen Abujaber, MD

Trinity Centre for Global Health, Trinity College Dublin

---

## [Reviewer Report]

Congratulations on a well prepared manuscript and optimising feedback from reviewers. No further changes suggested.

---

## [Reviewer Report]

Thank you for responding satisfactorily to my comments. This has improved the quality of your manuscript. This research will contribute to the general knowledge concerning gender and supportive supervision of MHPSS services in humanitarian contexts. I have no further comments. I wish you good luck with the next steps.